# FROM INDIVIDUAL TO MULTI-AGENT ALGORITHMIC RECOURSE: MINIMIZING THE WELFARE GAP VIA CAPACITATED BIPARTITE MATCHING

## ABSTRACT

Decision makers are increasingly relying on machine learning in sensitive situations. In such settings, *algorithmic recourse* aims to provide individuals with actionable and minimally costly steps to reverse unfavorable AI-driven decisions. While existing research predominantly focuses on single-individual (i.e., *seeker*) and single-model (i.e., *provider*) scenarios, real-world applications often involve multiple interacting stakeholders. Optimizing outcomes for seekers under an *individual welfare* approach overlooks the inherently multi-agent nature of real-world systems, where individuals interact and compete for limited resources. To address this, we introduce a novel framework for multi-agent algorithmic recourse that accounts for multiple recourse seekers and recourse providers. We model this many-to-many interaction as a *capacitated weighted bipartite matching problem*, where matches are guided by both recourse cost and provider capacity. Edge weights, reflecting recourse costs, are optimized for *social welfare* while quantifying the welfare gap between individual welfare and this collectively feasible outcome. We propose a three-layer optimization framework: (1) basic capacitated matching, (2) optimal capacity redistribution to minimize the welfare gap, and (3) cost-aware optimization balancing welfare maximization with capacity adjustment costs. Experimental validation on synthetic and real-world datasets demonstrates that our framework enables the many-to-many algorithmic recourse to achieve near-optimal welfare with minimum modification in system settings. This work extends algorithmic recourse from individual recommendations to system-level design, providing a tractable path toward higher social welfare while maintaining individual actionability.

## 1 INTRODUCTION

AI decision-making systems rapidly apply predictive models to support individuals in various contexts, e.g., loan approvals, medical treatments, or bail decisions (Voigt & Von Dem Bussche, 2017). The increasing reliance of humans on these algorithmic decision-making systems and their significant impact on areas such as finance, healthcare, and criminal justice have raised concerns regarding the transparency and fairness of these automated systems. Driven by AI policy regulations and the idea of a "right to explanation," algorithmic recourse is an emerging field that aims to provide individuals affected by negative, high-stakes algorithmic decisions with recommendations on how to reverse those outcomes (GDPR, 2016; Verma et al., 2024). Therefore, algorithmic recourse refers to the systematic process of reversing unfavorable decisions made by algorithms across various counterfactual scenarios (Wachter et al., 2018). It encompasses the necessary actions individuals must take to achieve a favorable outcome, serving as a foundation for temporally extended agency and trust in automated decision-making systems (Karimi et al., 2021). This concept is essential to ensure automated decisions are understandable and to enable individuals to engage with and contest these decisions (Doshi-Velez & Kim, 2017; Gunning, 2019). Existing studies on algorithmic recourse predominantly address how the individual would need to change their attributes to achieve the desired outcome (Karimi et al., 2022). Such settings generally assume a single individual impacted by a single decision-making model as shown in Figure 1a. In real-world scenarios, however, AI decision-making systems (i.e., *providers*) often interact with multiple individuals whose actions can

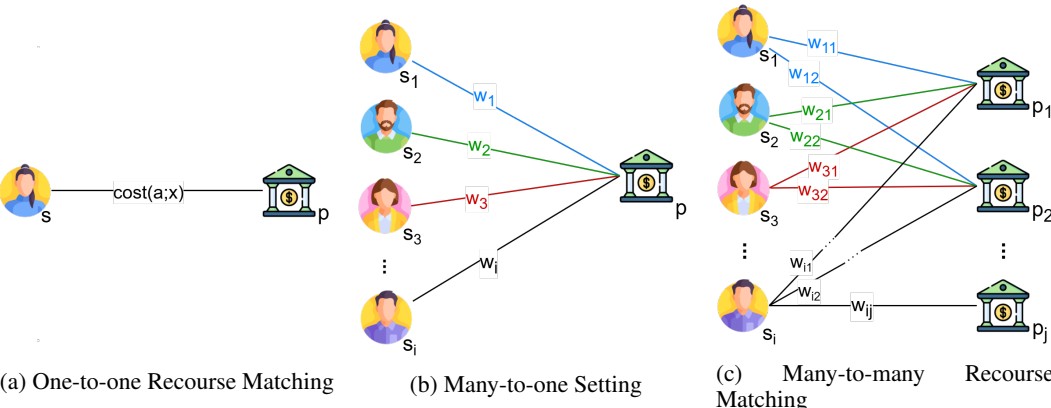

Figure 1: Various algorithmic recourse setups: (a) the classical one-to-one setting, where an individual $s$ seeks recourse recommendations from a provider $p$ with minimal cost required to reverse the output; (b) the many-to-one setting, where multiple individuals are seeking recourse from a single recourse provider. (c) Our proposed many-to-many framework generalizes prior settings by simultaneously optimizing for multiple recourse seekers and providers.

influence outcomes and, consequently, recourse recommendations for others (Many-to-one setting). Furthermore, individuals seeking recourse (i.e., *seekers*) may engage with multiple providers (Figure 1c) to choose the most suitable among given recommendations. The driving insight of this paper is that the effects of other stakeholders should not be ignored.

Recent studies have explored recourse frameworks that consider more than one seeker. For instance, O'Brien & Kim (2022) modified the recourse optimization problem, introducing Social-Welfare-Efficient Recourse and Pareto-Efficient Recourse. Drawing upon game-theoretic principles, their illustrative use of the prisoner's dilemma highlights potential misaligned benefits in conventional recourse recommendations and provides a compelling ethical rationale for reconsidering algorithmic interventions in a many-to-one setting. In similar vein, Fonseca et al. (2023) propose an agent-based simulation framework modeling multi-seeker-single-provider dynamic competition among recourse seekers over time, studying whether negatively classified instances will ever attain recourse in light of new individuals entering the seeker pool. Their work reveals that initially promising recourse recommendations may lose effectiveness due to continuous environmental changes. This emphasizes the importance of accounting for both competition and temporal shifts when formulating interventions. While these studies have extended the literature to settings with multiple recourse seekers, they continue to assume a single provider (Figure 1b). There remains a gap regarding algorithmic recourse in situations involving multiple recourse providers each potentially impacting outcomes with their own decision models. Existing approaches typically overlook how providing recourse recommendations simultaneously to multiple recourse seekers can benefit society and overall recourse actionability through interactions among individuals.

**Our Contributions** We extend existing recourse frameworks to move past the unrealistic assumption of infinite provider capacity wherein seekers are able to match with any provider in absence of other simultaneous matches. Our proposed formulation includes multiple recourse seekers and multiple recourse providers and examines how individual recommendations in such settings affect the overall system. We formalize this interaction as a *capacitated weighted bipartite matching problem* and determine optimal recourse outcomes using a linear-programming approach, thereby maximizing social welfare under capacity constraints. Specifically, we evaluate the population cost or social welfare of recourse, summed over all individuals, under realistic capacity limits and contrast it with the case of unlimited provider capacity.

Further, we identify a welfare gap between the socially optimal solution, computed by a central planner, and the unrealistic individually optimal outcome, where each seeker acts in isolation and selects the provider offering the lowest recourse cost, without coordination or consideration of provider capacity constraints. To minimize this gap, we introduce a second optimization layer that finds the

best distribution for a total fixed capacity over providers. We propose a systematic procedure to find this optimal distribution and minimize the gap.

Finally, since algorithmic recourse methods provide recommendations that minimally change the initial situation to reach the favourable outcome, we add the third optimization layer that minimizes the welfare gap while penalizing deviations from the initial capacity values. Solving this problem yields both the best provider capacities, accounting for the modification penalty, and the corresponding social welfare matching.

## 2 MANY-TO-MANY RECOURSE OPTIMIZATION

In this section, we formalize the matching problem that serves as the foundation of our optimization framework. The problem is modeled as a bipartite graph as shown in Figure 1c, with loan seekers on one set and loan providers (e.g., banks) on the other, as follows:

- **Seekers** $\mathcal{S}$: $\{s_i \mid s_i \in \mathcal{S}, \ \forall \ i \in [n]\}$, each characterized by a feature vector $x_i$.
- **Providers** $\mathcal{P}$: $\{p_j \mid p_j \in \mathcal{P}, \ \forall \ j \in [m]\}$, each equipped with a classifier (w.l.o.g. binary model) $h_j$ to accept or reject the seekers and a matching capacity $k_j$. All the seekers are initially rejected by all providers, i.e., $h_j(x_i) = -1 \ \forall \ i, j$, meaning that each seeker will have a recommendation from all providers.

Furthermore, it is assumed that a *central planner* will coordinate matches between seekers and providers (i.e., eq. (1)) and potentially redistribute existing capacity among providers (i.e., eq. (2) and eq. (3)).

**Recourse cost computation**   The standard algorithmic recourse setting assumes that a seeker (e.g., $s_i$) seeks to obtain recourse recommendations from a provider (e.g., $p_j$). This minimal change, as defined by Ustun et al. (2019), is defined as the minimal effort or cost required to change an individual's input features such that a predictive model will reverse its output from an undesirable outcome to a desirable one. Formally, given provider $p_j$'s decision model $h_j$ and an input feature vector $x_i$ corresponding to the characteristics of seeker $s_i$, such that $h_j(x_i) = -1$ (assumed binary w.l.o.g.), the recourse cost is defined as the solution to the following optimization problem:

$$c_{ij} = \min_{a \in A(x_i)} \text{cost}(a; x_i) \quad \text{s.t.} \quad h_j(x_i + a) = +1 \quad \forall \ i, j$$

where $a$ is an action vector representing feasible changes to the features of $x_i$, and $A(x_i)$ is the set of allowed actions based on domain constraints (e.g., mutability and bounds on feature changes). The function $\text{cost}(a; x_i)$ quantifies the difficulty of applying action $a$ to instance $x_i$. If a feasible action exists that satisfies the constraint, the minimal value of $\text{cost}(a; x_i)$ is the *recourse cost*. Therefore, $c_{ij}$ represents the minimal change required for seeker $s_i$ to achieve approval from provider $p_j$. Our proposed framework is agnostic to the choices of recourse method and providers' model, operating only on the minimum cost of change required for each pair of seeker $s_i$ and provider $p_j$.

**Bipartite Graph Construction.**   Once all minimal recourse costs, $c_{ij}$, between seekers and providers are precomputed, we construct a *weighted bipartite graph* $\mathcal{G} = (\mathcal{V}, \mathcal{W})$, where nodes $\mathcal{V} = \mathcal{S} \cup \mathcal{P}$ and

$$\mathcal{W} := \left\{ w_{ij} \mid w_{ij} = e^{-\gamma \cdot c_{ij}}, \ \forall \ i, j \right\}.$$

where $\gamma > 0$ is a scaling parameter controlling the sensitivity of the transformation. This exponential transformation converts costs into edge weights, enabling algorithms such as the maximum-weight bipartite matching (Kuhn, 1955) to prioritize low-cost (i.e., efficient) recourse assignments while maximizing overall match coverage. Furthermore, the exponential form ensures that differences among lower costs are emphasized more strongly than among higher ones, effectively prioritizing assignments that are not only feasible but also efficient.

**Optimization Model**   Next, with consideration for real-world constraints on provider capacity, we formulate a *capacitated weighted bipartite matching problem*. If we denote the maximum weight for seeker $i$ accordingly as $w_i^* = \max_j(w_{ij})$ (corresponding to the least costly recommendation

received), we can then measure the ideal scenario in which each seeker attains its optimal outcome, independently of other seekers, as:

$$\text{Individual Welfare} := \sum_{i=1}^{n} w_i^*$$

However, this ideal scenario assumes that providers have unbounded capacity (w.l.o.g., at least the number of seekers for each provider), meaning that they can freely provide the resources, which is not realistic. In practice, each provider has a limited capacity $k_j$, meaning they can serve only a finite number of seekers. Taking a systems-level view and aiming to minimize the overall cost of recourse across all seekers and providers,[1] the optimal matching under capacity constraints is determined as:

$$\text{Social Welfare} := \max_{z_{ij}} \sum_{i=1}^{n} \sum_{j=1}^{m} w_{ij} \, z_{ij}.$$

where $z_{ij}$ are binary decision variables that indicate whether seeker $i$ is assigned to provider $j$. To obtain the optimal matching, we encode the above formulation as a mixed-integer linear program (MILP) and solve it with the Gurobi Optimizer (Gurobi Optimization, LLC, 2024)[2] as follows:

$$\text{Social Welfare} := \max_{z_{ij}} \quad \sum_{i=1}^{n} \sum_{j=1}^{m} w_{ij} \, z_{ij}$$

$$\text{s.t.} \quad \underbrace{\sum_{j=1}^{m} z_{ij} \leq 1 \quad \forall i,}_{\text{Matching Constraint}} \quad \underbrace{\sum_{i=1}^{n} z_{ij} \leq k_j \quad \forall j,}_{\text{Capacity Constraint}} \quad \underbrace{z_{ij} \in \{0,1\} \quad \forall i,j}_{\text{Edge Constraint}} \quad (1)$$

Under unbounded provider capacity, the optimal matching over $z_{ij}$ is achieved when each seeker is matched with the provider that minimizes its individual recourse cost. The capacity constraints, however, may result in some seekers matching to a more costly match (lower weight $w_{ij}$). This discrepancy is quantified by the gap between the ideal individual welfare and the realized social welfare:

$$\text{Welfare Gap} := \left( \sum_{i=1}^{n} w_i^* \right) - \left( \sum_{i=1}^{n} \sum_{j=1}^{m} w_{ij} \, z_{ij} \right).$$

This gap highlights a critical design challenge: given a fixed total amount of provider capacity, how should these limited resources be distributed across providers to minimize the welfare gap? A naive uniform distribution of provider capacities may lead to significant welfare losses. In contrast, allocating more capacity to providers most preferred by seekers, i.e., those associated with lower recourse costs, can substantially reduce the welfare gap, even when the total capacity remains fixed. In the following section, we introduce a systematic approach that not only identifies distinct allocation scenarios based on varying resource availability but also provides an optimization approach to allocate capacity effectively and minimize the welfare gap.

## 3 MINIMIZE WELFARE GAP

Under a fixed total provider capacity $K = \sum_{j=1}^{m} k_j$, the welfare gap can vary depending on how capacity is distributed among providers. In fact, for any given $K$, there is an optimal allocation of provider capacities $k_j$ that minimizes this welfare gap. This observation leads to a new optimization problem involving two sets of decision variables namely, the integer variables $k_j \; \forall \, j$, representing provider capacities in the optimal solution, and the matching variables $z_{ij}$, as previously defined, indicating the best matching under the system settings.

$$\max_{z_{ij}, \, k_j} \quad \sum_{i=1}^{n} \sum_{j=1}^{m} w_{ij} \, z_{ij} \quad \text{s.t.} \quad \sum_{j=1}^{m} k_j = K \quad \forall j \quad \text{Total Capacity Constraint} \quad (2)$$

---

[1]This assumes that providers do not have ulterior preferences affecting the matching process.

[2]Although the presence of binary decision variables renders the problem NP-hard, Gurobi's branch-and-bound engine—augmented with presolve, cutting-plane generation, and heuristic warm-starts–guarantees global optimality.

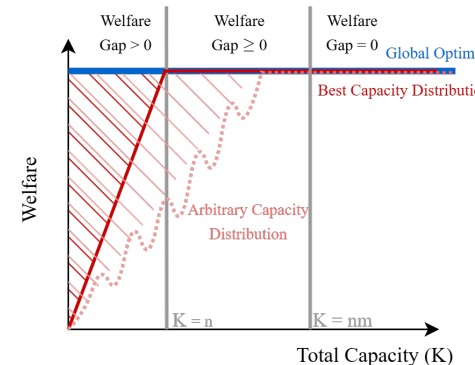

Figure 2: The effect of capacity distribution on social welfare outcomes. The pink dashed plot (with shaded region) shows the welfare achieved under an arbitrary capacity allocation (Equation (1)), which may result in a non-zero welfare gap even when sufficient total capacity is available. The dark red plot represents the optimized distribution obtained by solving Equation (2), which maximizes social welfare under a total capacity constraint. The blue horizontal line shows the global individual-optimal welfare, attainable only when total capacity is unconstrained. The gap between the pink and red curves highlights the inefficiency introduced by uncoordinated capacity allocations.

The matching, capacity, and edge constraints remain the same as before in Equation (1), with the additional constraint on the total capacity. Figure 2 represents how this welfare gap varies according to the total available resources and their distribution across providers. In particular, the figure highlights two notable cutoffs and three distinct areas. In the first area, where the total available resources are fewer than the number of seekers, i.e., $K = \sum_{j=1}^{m} k_j < |S|$, a welfare gap greater than zero is inevitable. Under these circumstances, some seekers will remain unmatched, and the depicted minimum gap represents the best achievable outcome through optimal capacity distribution. The first critical point is reached once available resources equal the number of seekers, where the welfare gap *can* reach zero *if* the distribution of these resources could be optimized and aligned with the best choice of each seeker. At the second critical point, where each provider individually possesses resources equal to the number of seekers, the welfare gap is guaranteed to be zero as seekers can freely match with their preferred providers. In the area between the two critical points, while resources are plentiful enough for a zero welfare gap distribution over capacities to exist, any suboptimal distribution of capacities among providers will result in a welfare gap greater than zero. Therefore, identifying the optimal capacity distribution becomes an essential challenge, specifically, determining how capacities can best be allocated to minimize the welfare gap given fixed total resources.

To determine the optimal distribution of provider capacities, Equation (2) could again be solved using a Mixed-Integer Linear Programming (MILP) method with solvers such as Gurobi (Gurobi Optimization, LLC, 2024). However, the structure of the matching weights reveals two key patterns, suggesting an approach simpler than solving an MILP directly. In Algorithm 1, we propose a systematic method that assigns capacities based on each seeker's top-ranked matching weight, previously defined in Section 2 as $w_i^*$. For each seeker $i$, the value $w_i^*$ represents the maximum possible contribution of that seeker to the total welfare, as no assignment can surpass this highest-weight edge. Moreover, since at most $K$ seekers can receive recourse, excluding any of the top $K$ highest-weight edges directly reduces the achievable welfare. Thus, the welfare of any feasible solution is bounded above by $\sum_{i \in \mathcal{S}_K} w_i^*$, where $\mathcal{S}_K$ denotes the set of seekers corresponding to the $K$ highest-ranked edges, and $j_i^* = \arg\max(w_i^*)$ denotes the index of the provider most preferred by seeker $s_i$. Then the capacity vector $k^*$ is defined by counting how often each provider appears among the top-K individually preferred matches (e.g. $w_i^*$).

$$k_j^* = |\{i \in S_K \mid j_i^* = j\}|, \quad j = 1, ..., m$$

These insights lead to Algorithm 1, which offers a more direct and efficient alternative to MILP. After computing all $w_i^*$ values in $O(nm)$ time ($n$ seekers and $m$ providers), we sort the resulting values once in $O(n \log(n))$ time. Capacities are then assigned to providers precisely according to

---

**Algorithm 1** Optimal Capacity Distribution

---

1: **Input:** seekers $\mathcal{S}$, providers $\mathcal{P}$, weights $w_{ij}$, total capacity $K$
2: **Output:** provider capacities $k = (k_1, \ldots, k_{|\mathcal{P}|})$
3: Initialize empty list $\mathcal{L}$
4: **for** each seeker $i \in \mathcal{S}$ **do**
5: $\quad$ $w_i^* = max_j(w_{ij})$
6: $\quad$ $j_i^* \leftarrow \arg\max_{j \in \mathcal{P}} w_i^*$ $\hspace{4cm}$ {best provider for seeker $i$}
7: $\quad$ Append triple $(i, \, j_i^*, \, w_i^*)$ to $\mathcal{L}$
8: **end for**
9: Sort $\mathcal{L}$ in descending order of weight
10: Select the first $K$ triples of $\mathcal{L}$ $\hspace{4cm}$ {top-$K$ matches}
11: Initialize $k_j \leftarrow 0$ for all $j \in \mathcal{P}$
12: **for** each selected triple $(i, \, j, \, w)$ **do**
13: $\quad$ $k_j \leftarrow k_j + 1$
14: **end for**
15: **return** capacity vector $k$

---

the top $K$ dominant edges. This approach ensures provider capacities align directly with seeker preferences, thereby naturally minimizing the welfare gap by emphasizing matches that are both individually optimal and beneficial for the overall system.

### 3.1 CAPACITY REDISTRIBUTION WITH PENALIZING MODIFICATIONS

While the previous section addressed how provider capacities can be optimally distributed to minimize the welfare gap for a given total system capacity $K$, additional considerations may be warranted. In practice, recourse methods often operate within established configurations determined by existing organizational structures, resource availability, and operational constraints. Transitioning from the current provider capacity configuration to an optimal setup typically involves real-world adjustment costs. Simply recommending an entirely new capacity distribution may be impractical or expensive to implement. Therefore, even though optimally reallocating capacities reduces the welfare gap, the costs of these adjustments must be carefully balanced against minimizing the welfare gap. To address this challenge, we extend our optimization framework by explicitly penalizing deviations from the initial capacities.

Let $\hat{k}_j$ denote the initial capacity of provider $j$, and $\tilde{k}_j$ represent the target capacity after configuration. The change in capacity $\Delta k_j = \tilde{k}_j - \hat{k}_j$ can penalize large changes with $\beta_j |\Delta k_j|$, where $\beta_j \geq 0$ controls the penalty sensitivity for each of the providers accordingly. Integrating this penalty into our optimization leads to a multi-objective problem, balancing social welfare maximization with minimization of capacity adjustment penalties. The modified objective function is:

$$\text{Welfare} = \max_{z_{ij}, \, k_j} \left( \sum_{i=1}^{n} \sum_{j=1}^{m} w_{ij} z_{ij} - \sum_{j=1}^{m} \beta_j |\Delta k_j| \right) \tag{3}$$

subject to the same matching, capacity, total capacity, and edge constraints previously defined in Equation (2). This enhanced formulation constitutes a MILP problem, solvable by recent versions of solvers like Gurobi (Gurobi Optimization, LLC, 2024). Ultimately, this optimization simultaneously identifies the optimal matching $z_{ij}$ and optimal provider capacities $k_j$, clearly indicating how capacities should be adjusted from their initial configuration. The resulting solution provides a practical recourse recommendation system effectively balancing improvements in social welfare with realistic operational constraints.

## 4 EXPERIMENTS

This section demonstrates how the proposed many-to-many recourse framework behaves in practice. All experiments are fully reproducible from the public Model-Agnostic Counterfactual Explanations (MACE) code base (Karimi et al., 2020) and the scripts accompanying this paper.

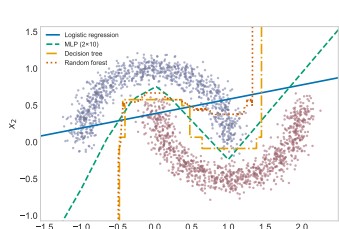

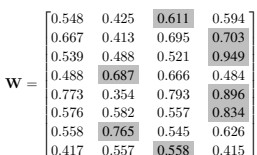

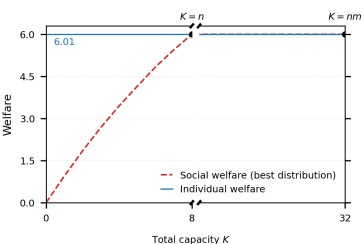

(a) Two-Moon dataset overlaid with the decision boundaries of the four provider models, underscoring that seekers face genuinely different models.

(b) Illustration of Algorithm 1 on the MACE-generated weight matrix. For each seeker (row), the largest edge weight $w_i^*$ is highlighted; its column index $j_i^*$ designates the preferred provider.

(c) Social welfare attained by Algorithm 1 (red dashed) as the total capacity $K$ grows from 0 to $n * m$, against the individual-welfare upper-bound.

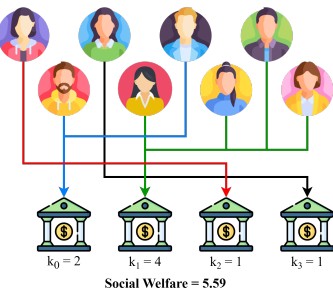

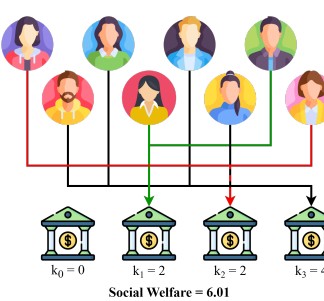

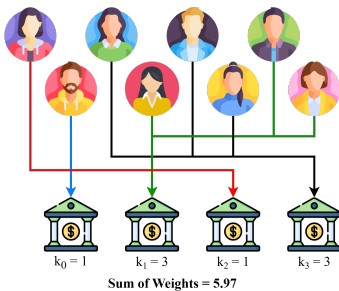

(d) Optimal seeker-to-provider assignments obtained by solving Equation (1) under the initial capacity vector.

(e) Provider capacities are distributed according to the capacity vector founded by Equation (2), eliminating the welfare gap.

(f) Final matching produced by Equation (3), balancing social-welfare gain against the cost of deviating from the initial capacities.

Figure 3: Comprehensive illustration of our proposed framework. (a) model decision boundaries, (b) weight matrix showing seeker-provider preferences and individually optimal matches, (c) welfare curve showing how total capacity impacts social welfare and its gap from the individual-welfare upper bound,(d) and the step-by-step optimization of provider capacities, (e) transitioning from initial assignments, to welfare-maximizing allocations, and finally (f) balancing welfare gains with capacity adjustment costs .

**Datasets and Models** We adopt three datasets, the synthetic Two-Moon benchmark, whose non-linear geometry yields heterogeneous classifiers with disagreements on decision boundaries, and the real-world COMPAS (Larson et al., 2016) and Credit (I-Cheng Yeh, 2009), widely used in recourse research. Four classification models serve as providers' models, namely, logistic regression, a multi-layer perceptron (MLP), a decision tree, and a random forest. After training the classification models on the datasets, those instances that were rejected by all providers are retained and the MACE code base is used to generate recourse actions for each pair of negatively predicted instance and classifier. We subsampled 8 seekers for Two-Moon, 12 for Credit, and 15 for COMPAS, forming the pools $\{s_i\}_{i=1}^{8}$, $\{s_i\}_{i=1}^{12}$, and $\{s_i\}_{i=1}^{15}$, respectively, used in our experiments. All panels in Figure 3 correspond to the Two-Moon dataset. We use this dataset to illustrate the full workflow of our framework due to its low dimensionality and visual interpretability. For real-world datasets, the overall patterns are similar, and results are reported in Table 1.

**Counterfactual Generation with MACE** Model-Agnostic Counterfactual Explanations (MACE) formulates counterfactual search as a constrained optimization problem, independent of the classifier(Karimi et al., 2020). It returns a minimal-distance counterfactual that flips the output to the favorable class while respecting feature mutability and domain bounds. In our experiments, we have used the $\ell_\infty$ norm as a measure of distance to identify the nearest (least costly) counterfactuals (results with the $\ell_1$ norm are included in the Appendix A.1). For every pair of seeker $s_i$ and provider $p_j$, MACE outputs a counterfactual instance $s_i^{CF}$ together with its recourse cost $c_{ij}$. These recourse cost values, 32 for Two-Moon (8 seekers × 4 providers), 48 for Credit (12 × 4), and 60 for COMPASS (15 × 4), constitute the primitive inputs to our framework. We ran MACE on a multi-CPU sys-

Table 1: Comparative results of the three optimization formulations. Each cell shows social welfare (SW), capacity vector $\mathbf{k}$, and the percentage of the individual-welfare upper bound (IW) attained in each optimization problem.

| | Two-Moon | | | Credit | | | COMPAS | | |
|---|---|---|---|---|---|---|---|---|---|
| | Capacity | SW | % IW | Capacity | SW | % IW | Capacity | SW | % IW |
| Equation (1) | $(2, 4, 1, 1)$ | 5.59 | 93.13% | $(3, 2, 6, 1)$ | 9.56 | 94.42% | $(3, 8, 1, 3)$ | 12.03 | 95.74% |
| Equation (2) | $(0, 2, 2, 4)$ | 6.01 | 100% | $(2, 0, 7, 3)$ | 10.12 | 100% | $(11, 1, 3, 0)$ | 12.57 | 100% |
| Equation (3) | $(1, 3, 1, 3)$ | 5.97 | 99.38% | $(3, 1, 6, 2)$ | 10.03 | 99.10% | $(5, 4, 3, 3)$ | 12.42 | 98.85% |

tem and will make the generated counterfactuals publicly available in our GitHub repository, which serves as the input to our framework. Furthermore, following Section 2, each cost is transformed into an edge weight by the exponential mapping. we have tried several values for gamma to indicate the differences between cost values well without altering their ordering, ensuring that low-cost recommendations dominate the subsequent matching and capacity allocation stages. Finally, we have chosen $\gamma = 10$ for the Two-Moon and COMPAS datasets and $\gamma = 600$ for the Credit dataset.

**Optimization Setup** The optimization proceeds in three layers. $(i)$ Given the total capacity $K$ equal to the number of seekers and an initial arbitrary capacity vector for each dataset, Equation (1) determines the maximum-weight matching. $(ii)$ For the total capacity $K$ increasing in range of $0, ..., n*m$ (though unrealistic), illustrating the full welfare curve and validate the theoretical cut-offs discussed in Section 2, Algorithm 1 finds the best distribution over capacities and the optimal value of the social welfare for the best matching in that setting. $(iii)$ Finally, the MILP approach solves Equation (3) and optimizes matching and capacities while penalizing deviations from $\hat{\mathbf{k}}$ with identical $\beta_j = 0.03$ for all providers.

**Results** Figure 3c illustrates the social welfare achieved with the capacity distribution returned by Algorithm 1 for each total capacity $K$ as it increases. The dashed red curve rises monotonically and meets the individual-welfare upper bound (solid blue) exactly at the point $K = n$. Beyond that, further capacity is non-essential, confirming the welfare gap would be zero in the case of finding the optimized distribution of capacities.

Figure 3d for Two-Moon dataset fixes $K = n$ (with $n = 8$) and starts from an arbitrary reference capacity vector $\hat{\mathbf{k}} = (2, 4, 1, 1)$; the first stage of our framework then computes the best matching for this initial configuration as shown in the figure. Although every seeker is matched, the allocation falls short of the global individual welfare. Figure 3b exposes the source of this gap. The next step of the algorithm finds each seeker's best edge $w_i^*$; distributing capacity according to Algorithm 1 and finds $(\mathbf{k}^* = (0, 2, 4, 4))$. With this distribution, the system reaches the welfare upper bound, which closes the gap entirely.

Finally, Figure 3f shows the outcome of the cost-aware optimization that penalizes deviations from $\hat{\mathbf{k}}$. The solver selects a softer move, $\tilde{\mathbf{k}} = (1, 3, 1, 3)$ yet the matching in this setting retains 99.38% of the global optimum. Hence, almost the full efficiency gain is achievable with limited system settings changes, demonstrating the practical value of our proposed framework.

To extend our evaluation, we summarize results for all datasets in Table 1, which highlights social welfare outcomes for each optimization setting and the corresponding optimal provider capacity distributions. For the Credit and COMPAS datasets, the best matching from Equation (1) attained a social welfare which constitutes 94.42% and 95.74% of the individual welfare, respectively. The cost-aware optimization in Equation (3) selects a softer redistribution that preserved 99.10% and 98.85% of the optimal welfare for Credit and COMPAS datasets while making a more balanced capacity modification, compared to the sharper concentration found in Equation (2). To the best of our knowledge, this is the first work that systematically aims to close the social welfare gap by moving beyond individual recourse recommendations and instead considering coordinated, system-level optimization across multiple seekers and providers. These results align with the findings from the synthetic dataset, reinforcing the effectiveness of targeted redistribution and demonstrating that even for complex, real-world data, high social welfare can be achieved with moderate system changes.

## 5 DISCUSSION

Our experiments show that welfare losses arise immediately in multi-agent settings when seekers apply independently under fixed, random capacities, even in a two-dimensional setting. However, these losses can be substantially reduced with minimal system-level intervention. When total system capacity matches the number of seekers, a single pass of Algorithm 1 restores the full social welfare. This suggests that most of the efficiency gap stems not from resource scarcity but from poor allocation. Moreover, introducing a modest penalty for deviating from the initial capacity distribution (Equation (3)) had little impact on overall performance. This indicates that near-optimal outcomes are achievable through small, targeted adjustments, supporting the practical feasibility of implementation in real-world systems.

The results also highlight the importance of model diversity. Providers whose classifiers align well with the geometry of a particular sub-population accrue many high-weight edges. Directing additional capacity to such specialists both increases overall welfare and improves individual actionability. In contrast, giving every provider the same capacity can waste resources on providers that do not add much value.

The third optimization Equation (3) introduced in Section 3.1 subsumes the previous two as special cases through the choice of the penalty coefficient $\beta_j$. When $\beta_j = 0 \quad \forall j$, the penalty term disappears, and the optimization reduces to maximizing social welfare without regard for deviations from the initial capacity vector $\hat{k}$; this is equivalent to the Equation (2), which computes the matching under best distribution of capacities. Conversely, when $\beta_i \to \infty \quad \forall j$, any deviation $\Delta k_j$ from the initial capacity incurs an infinite cost, effectively forcing $\Delta k_j = 0$ for all providers. This constraint recovers the Equation (1), which maximizes welfare subject to fixed initial capacities. Thus, the parameter $\beta_j$ enables interpolation between these two extremes, offering a flexible mechanism to balance social welfare against capacity adjustments.

While our experimental results demonstrate the effectiveness of the proposed framework across different datasets and settings, it is important to note that several design choices can influence the specific quantitative outcomes observed in figures and tables. These include the choice of the edge-weight transformation function, the scaling parameter $\gamma$, and the penalty coefficients $\beta_j$ used in cost-aware optimization. However, the central insight remains robust, extending the recourse framework from a one-to-one setting to a many-to-many setting and studying its effects. Looking forward, this framework can be enriched by considering game-theoretic extensions in which providers act strategically. Such formulations could capture competitive or decentralized recourse environments, opening new directions for modeling strategic behavior and fairness in multi-agent recourse systems.

## 6 CONCLUSION

We have introduced a many-to-many view of algorithmic recourse in which multiple seekers obtain recommendations from multiple decision-making models whose resources are limited. Further, we quantified the welfare gap between the socially optimal solution, computed by a central planner, and the individually optimal outcome, where each seeker acts in isolation and selects the provider offering the lowest recourse cost, without coordination. Experiments demonstrated that explicit capacity management improves social welfare to almost 99% of the theoretical upper bound while modifying minor capacity units.

Beyond the static setting studied here, several adjustment levers deserve closer attention. On the seeker side, adding or removing applicants or allowing limited edits to their feature vectors directly reshapes the social welfare landscape. On the provider side, changing the number of providers, altering classifier parameters transforms both the attainable optimum and the route toward it. Future studies can link these adjustment options to policy goals like ensuring diversity, fairness, or profit. A second line of extension is dynamism. Real recourse ecosystems are not one-shot games; seekers reapply, models retrain, and resources fluctuate. Embedding the matching layer inside an online setting would let the system adapt capacities and decision boundaries in real time, continuously shrinking the welfare gap as new data arrive. overall, the proposed framework bridges the space between individual prescriptions and system-wide outcomes, offers a tractable path toward higher social welfare, and opens several promising directions for adaptive and fairness-aware extensions.

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

# A APPENDIX

## A.1 ADDITIONAL EXPERIMENTS WITH $\ell_1$ DISTANCE

To test robustness to the distance function, we repeated all experiments with the $\ell_1$ norm instead of the $\ell_\infty$ norm. The datasets, provider models, and optimization procedure were unchanged, except for minor parameter adjustments. For Two-Moon, we used 10 seekers with $\gamma = 20$ and $\beta = 0.02$; for Credit and COMPAS, we kept the setup and $\gamma$ but increased $\beta$ to 0.05. As shown in Figure 4, the framework still achieves near-optimal welfare with minimal adjustments, and quantitative results for all datasets are summarized in Table 2.

$$\mathbf{W} = \begin{bmatrix} 0.256 & 0.104 & 0.256 & \boxed{0.353} \\ 0.383 & 0.096 & 0.366 & \boxed{0.441} \\ 0.179 & 0.148 & 0.129 & \boxed{0.902} \\ 0.415 & 0.064 & 0.432 & \boxed{0.800} \\ 0.187 & 0.237 & 0.168 & \boxed{0.694} \\ 0.299 & \boxed{0.499} & 0.178 & 0.393 \\ 0.165 & \boxed{0.214} & 0.198 & 0.171 \\ 0.211 & 0.021 & 0.230 & \boxed{0.251} \\ 0.509 & 0.049 & 0.535 & \boxed{0.727} \\ 0.368 & \boxed{0.830} & 0.314 & 0.395 \end{bmatrix}$$

(a) Illustration of Algorithm 1 on the MACE-generated weight matrix. For each seeker (row), the largest edge weight $w_i^*$ is highlighted; its column index $j_i^*$ designates the preferred provider.

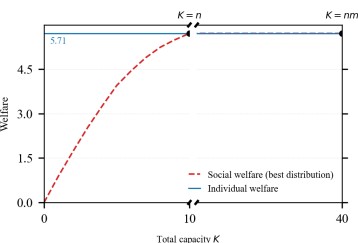

(b) Social welfare attained by Algorithm 1 (red dashed) as the total capacity K grows from 0 to $n*m$, plotted against the individual-welfare upper-bound.

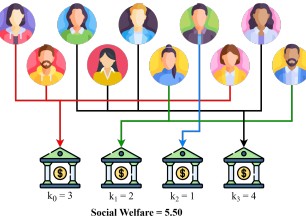

(c) Optimal seeker-to-provider assignments obtained by solving Equation (1) under the initial capacity vector.

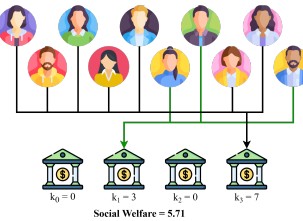

(d) Provider capacities are distributed according to the capacity vector founded by Equation (2), eliminating the welfare gap.

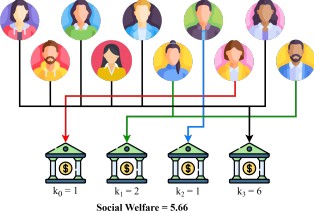

(e) Final matching produced by Equation (3), balancing social-welfare gain against the cost of deviating from the initial capacities.

Figure 4: Comprehensive illustration of our proposed framework. This figure demonstrates the complete process: weight matrix showing seeker-provider preferences and individually optimal matches (a), welfare curve showing how total capacity impacts social welfare and its gap from the individual-welfare upper bound (b), and the step-by-step optimization of provider capacities, transitioning from initial assignments (c), to welfare-maximizing allocations (d), and finally balancing welfare gains with capacity adjustment costs (d)

Table 2: Comparative results of the three optimization formulations. Each cell shows social welfare (SW), capacity vector $\mathbf{k}$, and the percentage of the individual-welfare upper bound (IW) attained in each optimization problem.

| | **Two-Moon** | | | **Credit** | | | **COMPAS** | | |
|---|---|---|---|---|---|---|---|---|---|
| | Capacity | SW | % IW | Capacity | SW | % IW | Capacity | SW | % IW |
| Equation (1) | $(3, 2, 1, 4)$ | 5.50 | 96.32% | $(3, 2, 6, 1)$ | 7.49 | 87.04% | $(3, 8, 1, 3)$ | 12.03 | 95.74% |
| Equation (2) | $(0, 3, 0, 7)$ | 5.71 | 100% | $(4, 0, 3, 5)$ | 8.61 | 100% | $(11, 1, 3, 0)$ | 12.57 | 100% |
| Equation (3) | $(1, 2, 1, 6)$ | 5.66 | 99.03% | $(3, 0, 6, 3)$ | 8.44 | 98.03% | $(5, 6, 1, 3)$ | 12.26 | 97.53% |

THE USE OF LARGE LANGUAGE MODELS

Large Language Models were used in a limited manner to improve grammar and polish the writing in certain sections of the manuscript. They were also employed to assist with retrieval and discovery, such as identifying and summarizing related work.

