# OpenReview forum: "From Individual to Multi-Agent Algorithmic Recourse: Minimizing the Welfare Gap via Capacitated Bipartite Matching"
_ICLR.cc/2026/Conference — ICLR 2026 Conference Withdrawn Submission_

### Official Review · Reviewer_3cbM · 2025-10-29

**Soundness:** 3
**Presentation:** 3
**Contribution:** 2
**Rating:** 2
**Confidence:** 2

**Summary:**

This paper studies the "many-to-many" recourse optimization problem where traditional model assumes a one-to-one structure. It models the system as a capacitated weighted bipartite matching problem. They introduce the problem as a mixed-integer linear programming with capacity constraints. Experiments show this final approach achieves near-optimal social welfare with minimal changes to the system.

**Strengths:**

1. The many-to-many model is well-motivated as a realistic scenario in real-world applications.

2. The three-layer optimization framework is well presented and experiments support the claim.

**Weaknesses:**

1. While the paper addresses an important and practical problem in recourse optimization, it lacks theoretical depth which is below the ICLR bar. The problem is formulated as a mixed-integer linear programming (eq. 1) and then is solved through existing approaches. The contribution of this work is more of a problem formulation, which is the main weakness.

2. The paper models the recourse problem as a single, static matching event. Real-world systems are dynamic: new seekers enter the pool, seekers re-apply, and providers retrain their models. The current framework does not address how the system would adapt to these changes.

**Questions:**

1. Have you considered fairness objectives?

2. What is the computational cost of the solution? Does it scale with the market size?

---

> ### Author Response · Authors · 2025-11-22
> **Review-3cbM — Computational Cost and Scalability, Static Setting, and Fairness Objectives**
>
> Review-3cbM — Computational Cost and Scalability.
>
> Eq.(1) has the structure of a classical capacitated bipartite matching problem, solvable in polynomial time via min-cost-flow/assignment algorithms. Eq.(2) introduces integer capacity variables, a capacity vector $k$ with $\sum_{j} k_j = K$ and the associated matching. Naively, this would require searching over a large discrete set of capacity allocations, whereas Algorithm1 provides a direct, closed-form, polynomial-time procedure: it computes all $w_{ij}^\*$ in $O(nm)$, sorts them in $O(n \log n)$, and selects the top-$K$ edges to obtain the welfare-maximizing capacity distribution. Algorithm1 recovers the same optimal social welfare as solving Eq.(2) with Gurobi; we therefore use it to efficiently sweep over different values of $K$ and generate the red welfare curve in Fig.3c. Eq.(3) is no longer the standard matching/flow structure that makes Eq.(1) polynomial-time solvable, so it cannot be handled by classical matching algorithms and is instead formulated and solved as a mixed-integer program. So, regarding scalability, Layers 1 and 2 scale polynomially in the number of seeker and provider pairs, so they can handle substantially larger markets. Layer 3 operates on provider-level capacities as a mixed-integer problem and is inexpensive at the moderate market sizes we consider; for very large markets, it can be applied to a reduced set of high-impact providers.
>
> Review-3cbM – Static Setting and Fairness Objectives.
>
> The framework is static because it directly extends the basic static recourse model; we agree that dynamic systems are important and have been discussed in Section 6. We do not optimize fairness objectives explicitly here, our goal is to study social welfare and feasibility of minimal-cost recourse under capacity constraints, but as we have also mentioned in Section 6, future extensions could incorporate fairness via group-weighted utilities or group-level constraints. However, we will add this paragraph to Section 3 to illustrate how our welfare gap aligns with related works in this area "Our notion of a welfare gap is conceptually aligned with prior work in welfare-based fairness and allocation. Research on welfare-aware decision making (e.g.,Hu & Chen ,2020; Heidari et al., 2019; Binns, 2018) analyzes how system-level welfare changes when constraints or fairness criteria are included, while work on allocation and distributive justice (Rea et al., 2021; Cousins et al., 2023) studies how limited resources should be assigned to maximize aggregate benefit under capacity or harm constraints. More recent studies (Perello et al., 2025), highlight how recourse objectives themselves may raise disparities, framing recourse as a resource that must be allocated fairly. Our welfare-gap metric plays a similar evaluative role, it quantifies the loss relative to a normative unconstrained benchmark, capturing how capacity constraints reduce achievable social welfare."
> We appreciate the reviewer’s "Good" assessments of soundness and presentation, and in light of these clarifications about the scope and contribution of the work, we respectfully ask the reviewer to reconsider their overall recommendation.
>
> REFERENCES
>
> [1] Hu, L., & Chen, Y. (2020, January). Fair classification and social welfare. In Proceedings of the 2020 conference on fairness, accountability, and transparency (pp. 535-545).
>
> [2] Heidari, H., Ferrari, C., Gummadi, K., & Krause, A. (2018). Fairness behind a veil of ignorance: A welfare analysis for automated decision making. Advances in neural information processing systems, 31.
>
> [3] Fairness in machine learning: Lessons from political philosophy. Proceedings of the 2018 Conference on Fairness, Accountability, and Transparency (FAT*).
>
> [4] Rea, D., Froehle, C., Masterson, S., Stettler, B., Fermann, G., & Pancioli, A. (2021). Unequal but fair: Incorporating distributive justice in operational allocation models. Production and Operations Management, 30(7), 2304-2320.
>
> [5] Cousins, C., Viswanathan, V., & Zick, Y. (2023, December). The good, the bad and the submodular: Fairly allocating mixed manna under order-neutral submodular preferences. In International Conference on Web and Internet Economics (pp. 207-224). Cham: Springer Nature Switzerland.
>
> [6] Perello, N., Cousins, C., Zick, Y., & Grabowicz, P. (2025, June). Discrimination Induced by Algorithmic Recourse Objectives. In Proceedings of the 2025 ACM Conference on Fairness, Accountability, and Transparency (pp. 1653-1663).

---

> > ### Comment · Reviewer_3cbM · 2025-11-26
> >
> > Thank you for your response. While the authors answered my questions, the novelty and contribution of this submission still looks limited to me. Therefore, I decide to maintain the score while keeping the low confidence.

---

### Official Review · Reviewer_LPBC · 2025-11-01

**Soundness:** 1
**Presentation:** 2
**Contribution:** 1
**Rating:** 0
**Confidence:** 4

**Summary:**

The paper considers an algorithmic recourse problem: individuals that are rejected by a classification models are provided some recourse information that corresponds to a minimum-cost change in their input to the model that would change the classifier's output to be positive. The objective is to assign the individuals to the classifiers respecting the classifier capacities minimizing the total cost. They also consider a variant where the capacities of the classifiers are variable and have an associated cost to increase.

**Strengths:**

The authors state that the scenario with multiple individuals seeking to be matched to one of many classifiers has not been considered before. I am not familiar with the algorithmic recourse literature specifically, but this seems to be the main strength of their paper.

**Weaknesses:**

The problems the authors solve are too simple to be of interest to a wider audience. The authors seem to be under the impression that the problems they are solving are NP-Hard. See for example, line 176, footnote 2. This is not true, since the matching polytope is integral. Any simple LP/flow solver suffices to solve LP (1). The remainder of their problems are also not NP-Hard and can be easily solved exactly using simple known techniques, and do not require heavy machinery like Gurobi.

Algorithm 1 to solve LP (2) is also not very interesting, since it corresponds to the obvious policy of assigning each seeker to its most preferred (ie least-costly) provider.

LP (3) is also solving a polynomial-time solvable problem. It can be reformulated as finding a min-cost flow. Each seeker can be represented as a node with an edge of capacity 1 from the source. Each seeker has edges to nodes corresponding to providers. The edge from seeker i to provider j has capacity 1 and cost -w_ij. The node corresponding to provider j has two edges to the sink. The first edge has cost 0 and capacity k_j. The second edge has infinite capacity and cost \beta_j. The min cost flow precisely solves the LP.

The instances the authors solve experimentally are very small, perhaps because they are using complex solvers. But as I mentioned, this is not necessary.

**Questions:**

Capacity augmentation has seen some more work recently. I suggest the authors look up some of the literature on this or flexible capacities.

---

> ### Author Response · Authors · 2025-11-22
> **Review-LPBC – Complexity of the Problem and Contribution.**
>
> Review-LPBC – Complexity of the Problem and Contribution.
>
> Our intention in the footnote was not to claim that our formulations are NP-hard, but to justify using a general-purpose MILP solver such as Gurobi for problems with integer decision variables; we will revise the wording to avoid this misunderstanding. Eq.(3) differs from the min-cost-flow construction suggested in the review because capacities $k_j$ are decision variables and we penalize deviations from an initial vector $\hat{k}$, so the model must jointly choose assignments and capacities; for this reason we formulate it as a mixed-integer optimization problem. Finally, we would like to point out that the main contribution of our work lies in formulating many-to-many algorithmic recourse as a capacitated bipartite matching problem with a welfare-gap analysis, an aspect the reviewer noted as a strength, rather than in algorithmic hardness. we respectfully believe that this misunderstanding may have influenced the overall recommendation, and we kindly ask the reviewer to reconsider in light of these clarifications.

---

### Official Review · Reviewer_tY72 · 2025-11-02

**Soundness:** 3
**Presentation:** 3
**Contribution:** 2
**Rating:** 4
**Confidence:** 3

**Summary:**

This paper proposes extending algorithmic recourse from individual to multi-agent settings by framing the problem as a capacitated, weighted bipartite matching between recourse seekers and recourse providers. The authors define a three-layer optimization framework that (1) performs welfare-maximizing matching, (2) redistributes provider capacities to minimize the gap between individual and social welfare, and (3) regularizes capacity adjustments to balance welfare gains against deviation costs. The method is evaluated on synthetic and real-world data and is shown to improve overall social welfare under capacity constraints.

**Strengths:**

Addresses a meaningful and underexplored problem: multi-agent aspects of algorithmic recourse.

Formulates a clear optimization approach grounded in established matching theory.

The layered structure (matching → redistribution → penalized adjustment) is logically organized and tractable.

Empirical results are consistent with the theoretical claims and provide useful illustration of welfare trade-offs.

**Weaknesses:**

**Conceptual misframing and motivation**
The main weakness lies in the conceptual grounding of the work. The paper presents itself as advancing algorithmic recourse, but the formulation and experiments primarily concern resource allocation under capacity constraints. The usual meaning of recourse involves providing actionable interventions to reverse an algorithmic decision (e.g., a classifier output), but no such model or decision process appears here. The “providers” do not represent decision-making systems, and the “recourse costs” are exogenous edge weights. As a result, the setting is more naturally understood as a fair allocation or matching problem rather than an extension of algorithmic recourse. This isn't disqualifying in and of itself, but this connection should be explored. The connection to recourse theory, feature-space manipulation, or model-driven feedback is not clearly justified.

**Modeling incoherence**
Even within the proposed framework, several modeling assumptions are underspecified. The recourse costs $c_{ij}$ are treated as known, fixed, and comparable across seekers and providers, but it is unclear how they are derived or whether they depend on the decision model, individual features, or other agents’ actions. The “individual optimum” (where seekers independently choose the lowest-cost provider) is not well-defined behaviorally—it assumes perfect information and ignores strategic effects, making the “welfare gap” somewhat contrived. The subsequent “capacity redistribution” layers appear as planner-side postprocessing, not something interpretable as policy or learning dynamics.

**Limited engagement with relevant literature**
The paper primarily cites standard recourse and interpretability works (e.g., Karimi et al., 2021; O’Brien & Kim, 2022) but omits key literatures that directly address the welfare and allocation aspects central to this formulation. In particular, perhaps the most relevant work beyond the above is not cited:

Perello, N., Cousins, C., Zick, Y., & Grabowicz, P. (2025, June). Discrimination Induced by Algorithmic Recourse Objectives. In Proceedings of the 2025 ACM Conference on Fairness, Accountability, and Transparency (pp. 1653-1663).


Fair and welfare-aware allocation. Some also handle harms, which is crucial in the recourse setting:

Rea et al. (2021) — Unequal but fair: Incorporating distributive justice in operational allocation models.

Cousins, Viswanathan, & Zick (2023) — The good, the bad, and the submodular.



Welfare-Based and Allocation-Oriented Learning:

The paper discusses “social welfare,” but don’t connect to the established literature on welfare-based optimization or multi-agent fairness in learning. This literature should be discussed, starting with, e.g.,

Lily Hu and Yiling Chen. 2020. Fair classification and social welfare. In Proceedings of the 2020 ACM Conference on Fairness, Accountability and Transparency (FAccT).

Binns, R. (2018). Fairness in machine learning: Lessons from political philosophy. Proceedings of the 2018 Conference on Fairness, Accountability, and Transparency (FAT*).

Heidari, H., Ferrari, C., Gummadi, K. P., & Krause, A. (2019). Fairness behind a veil of ignorance: A welfare analysis for automated decision making. NeurIPS 2019.

Kim, M. P., Ghorbani, A., & Zou, J. Y. (2019). Multiaccuracy: Black-box post-processing for fairness in classification. AAAI 2019. (Useful because it’s another welfare-related aggregation view on fairness.)

These connect directly to their welfare gap metric, and could justify that conceptual move more rigorously.

For the dynamics of interactive systems, other work explores sequential tasks, which also seem relevant here:

Cousins, Asadi, Lobo, & Littman (2024). On welfare-centric fair reinforcement learning (Reinforcement Learning Conference).
(Addresses recurrent decision settings where welfare and resource allocation interact dynamically.)




*Clarity and exposition*
The exposition could be improved. It is often unclear whether $c_{ij}$ values and recourse costs are known or estimated, and what their behavioral interpretation is. The figures do not clarify how the transition from one-to-one to many-to-many settings changes the mathematical problem. The optimization layers are described as algorithmic innovations, but they largely restate standard LP formulations.

Marginal technical novelty
The technical content—capacitated matching with welfare regularization—is standard in operations research and computational social choice. While applying it to “recourse” is a creative idea, the paper does not introduce new algorithms or theoretical results. The empirical results confirm expected monotonic trends (e.g., increasing capacity yields higher welfare), without surprising or interpretive insight.



Minor Issues:

043: Fix your quotation marks: a ”right to explanation.”

399: Please use math mode for math.

**Questions:**

Interpretation of “recourse”:
Could you clarify what constitutes recourse in your setting? Are the costs $c_{ij}$ derived from underlying model-based counterfactuals (as in standard recourse), or are they abstract utilities within a matching framework? If the latter, how does this remain within the conceptual scope of algorithmic recourse?

Knowledge and estimation of $c_{ij}$ and costs:
Do the seekers or providers know their corresponding $c_{ij}$ values, costs, or utilities? Are these assumed known to a central planner, or estimated empirically? How sensitive are your conclusions to uncertainty or estimation error in these quantities?

---

> ### Author Response · Authors · 2025-11-22
> **Review-tY72 – Conceptual framing and Motivation, Modeling Assumptions, Related Work and Scope, and Exposition and Empirical Results**
>
> Review-tY72 – Conceptual framing and Motivation.
>
> Although the connection of our approach to fairness and resource allocation is natural and worth exploring further, our primary aim here is to advance algorithmic recourse in a many-to-many setting. As you note, recourse refers to actionable interventions that flip a model’s decision; traditional single-to-single recourse recommendations do not consider the effect of multiple seekers and providers and capacity limits, so a minimum-cost intervention may not remain actionable or valid once many seekers target the same provider. This motivates our first layer (Eq.1), which, given individual recourse costs, computes a socially optimal matching that results in system-level actionable recourse, even if it is no longer individually optimal for every seeker.
>
> Review-tY72 - Modeling Assumptions
>
> Section 2 clearly defines $c_{ij}$ as a recourse cost derived from provider $p_j$'s prediction model and seeker $s_i$'s features, and we assume a central planner that has access to these costs and uses them to coordinate matches and capacities. The "individual optimum" is intended as an upper bound under unconstrained capacities, not a behavioral model; the welfare gap measures how far feasible allocations under capacity constraints fall below this benchmark.
>
> Review-tY72 – Related Work and Scope
>
> We agree that our framework can also be studied from a fairness perspective and connected more explicitly to welfare and allocation literature, and we will add a subset of the suggested references and this paragraph to Section~3 to address your comment "Our notion of a welfare gap is conceptually aligned with prior work in welfare-based fairness and allocation. Research on welfare-aware decision making (e.g.,Hu & Chen ,2020; Heidari et al., 2019; Binns, 2018) analyzes how system-level welfare changes when constraints or fairness criteria are included, while work on allocation and distributive justice (Rea et al., 2021; Cousins et al., 2023) studies how limited resources should be assigned to maximize aggregate benefit under capacity or harm constraints. More recent studies (Perello et al., 2025), highlight how recourse objectives themselves may raise disparities, framing recourse as a resource that must be allocated fairly. Our welfare-gap metric plays a similar evaluative role, it quantifies the loss relative to a normative unconstrained benchmark, capturing how capacity constraints reduce achievable social welfare."  However, our current goal is not to propose new fairness criteria, but to study whether the basic minimum cost required to obtain a favorable outcome for multiple seekers and providers can be made socially beneficial and actionable under capacity constraints.
>
> Review-tY72 - Exposition and Empirical Results.
>
> We already state that $c_{ij}$ represents the minimal change required for seeker $s_i$ to achieve approval from provider $p_j$, and our framework is agnostic to the choice of recourse method and providers’ model; any method that computes such costs can be plugged in, and we can make this point more prominent. Regarding the empirical results, our Layers 2 and 3 redistribute a fixed total capacity $K$, and in our experiments, for each fixed $K$, the improvements in social welfare arise from redistribution of capacity rather than simply increasing total capacity. Also, the minor issues will be fixed as well.
>
>
> REFERENCES
>
> [1] Hu, L., & Chen, Y. (2020, January). Fair classification and social welfare. In Proceedings of the 2020 conference on fairness, accountability, and transparency (pp. 535-545).
>
> [2] Heidari, H., Ferrari, C., Gummadi, K., & Krause, A. (2018). Fairness behind a veil of ignorance: A welfare analysis for automated decision making. Advances in neural information processing systems, 31.
>
> [3] Fairness in machine learning: Lessons from political philosophy. Proceedings of the 2018
> Conference on Fairness, Accountability, and Transparency (FAT*).
>
> [4] Rea, D., Froehle, C., Masterson, S., Stettler, B., Fermann, G., & Pancioli, A. (2021). Unequal but fair: Incorporating distributive justice in operational allocation models. Production and Operations Management, 30(7), 2304-2320.
>
> [5] Cousins, C., Viswanathan, V., & Zick, Y. (2023, December). The good, the bad and the submodular: Fairly allocating mixed manna under order-neutral submodular preferences. In International Conference on Web and Internet Economics (pp. 207-224). Cham: Springer Nature Switzerland.
>
> [6] Perello, N., Cousins, C., Zick, Y., & Grabowicz, P. (2025, June). Discrimination Induced by Algorithmic Recourse Objectives. In Proceedings of the 2025 ACM Conference on Fairness, Accountability, and Transparency (pp. 1653-1663).

---

### Official Review · Reviewer_uXJx · 2025-11-03

**Soundness:** 3
**Presentation:** 3
**Contribution:** 3
**Rating:** 6
**Confidence:** 4

**Summary:**

The paper introduces the problem of multi-agent algorithm recourse, formalized as a matching problem between recourse seekers and providers (e.g., the bank). The authors provide three different “flavors” of the optimization problem by casting it as an MILP problem (Equations 1, 2, and 3), and by also providing a custom algorithm. Lastly, the authors performed some experimental evaluation of their approach, under a synthetic and a real-world setting, showing the effectiveness of their approach on the welfare gap and related metrics.

**Strengths:**

I think the paper provides an interesting and original contribution to the field of algorithmic recourse. The paper is well-written, and the problem (and related solutions) are described nicely. In particular, I think the angle presented in Section 3.1 is very interesting, since it shows how sometimes it is the provider that has to adjust its parameters to improve recourse chances, fairness, and social welfare (as underlined by some papers [1,2,3]).

[1] Barrainkua, A., De Toni, G., Lozano, J. A., & Quadrianto, N. (2025). Who Pays for Fairness? Rethinking Recourse under Social Burden. arXiv preprint arXiv:2509.04128.

[2] Alejandro Kuratomi, Evaggelia Pitoura, Panagiotis Papapetrou, Tony Lindgren, and Panayiotis Tsaparas. Measuring the burden of (un) fairness using counterfactuals. In Joint European conference on machine learning and knowledge discovery in databases, pages 402–417. Springer, 2022.

[3] Francesca ED Raimondi, Andrew R Lawrence, and Hana Chockler. Equality of effort via algorithmic recourse. arXiv preprint arXiv:2211.11892, 2022.

**Weaknesses:**

I feel the theoretical tractability of the (bipartite matching) recourse problem is somewhat not well-explored. Indeed, besides relying on Gurobi, it would have been nice to see some analysis on the complexity of each “flavor” and/or potential approximation results (e.g., about Algorithm 1). For example, Equation 2 can be solved by Gurobi (which is quite efficient already). Thus, the relevance of Algorithm 1 is not clear not me, since in practice I might as well use the solver directly. It would have been nice to see some analysis of the approximation factor we get on the welfare gap with respect to Gurobi.

**Questions:**

- Can you better describe the relevance and/or theoretical improvements of Algorithm 1 with respect to the solution provided by the solver? For example, a comparison between Algorithm 1 and Gurobi produced gaps in Figure 3c?

---

> ### Author Response · Authors · 2025-11-22
> **Review-uXJx — Tractability and Algorithm 1**
>
> Eq.(1) has the structure of a classical capacitated bipartite matching problem, solvable in polynomial time via min-cost-flow/assignment algorithms. Eq.(2) introduces integer capacity variables, a capacity vector $k$ with $\sum_{j} k_j = K$ and the associated matching. Naively, this would require searching over a large discrete set of capacity allocations, whereas Algorithm1 provides a direct, closed-form, polynomial-time procedure: it computes all $w_{ij}^\*$ in $O(nm)$, sorts them in $O(n \log n)$, and selects the top-$K$ edges to obtain the welfare-maximizing capacity distribution. Algorithm1 recovers the same optimal social welfare as solving Eq.(2) with Gurobi; we therefore use it to efficiently sweep over different values of $K$ and generate the red welfare curve in Fig.3c. Eq.(3) is no longer the standard matching/flow structure that makes Eq.(1) polynomial-time solvable, so it cannot be handled by classical matching algorithms and is instead formulated and solved as a mixed-integer program.

---

> > ### Comment · Reviewer_uXJx · 2025-11-27
> >
> > I would like to thank the authors for their comments and clarifications. I have no further questions to ask. After reading the other reviews, I am inclined to maintain my score. Although I recognize the novelty of the contribution, I also believe the strength of this paper lies more in pushing a new problem setting than in the technical contribution per se, as some reviewers have already argued (LPBC).

---

### Author Response · Authors · 2025-11-22
**General Response**

We thank the reviewers for their thoughtful feedback and for their positive remarks. Reviewers agree that our paper addresses a meaningful, important, and previously unexplored multi-agent recourse setting (Reviewer uXJx, Reviewer tY72, Reviewer LPBC, Reviewer 3cbM), is well written and clearly presented (Reviewer uXJx, Reviewer tY72, Reviewer 3cbM), and proposes a well-motivated, realistic many-to-many model (Reviewer 3cbM). They found our optimization framework clear, well-structured, and grounded in matching theory (Reviewer tY72, Reviewer 3cbM), and highlighted the novel provider-side perspective we introduce (Reviewer uXJx).  Reviewers further noted that our experiments are consistent with theory (Reviewer tY72) and demonstrate effective, near-optimal improvements in social welfare with minimal system changes (Reviewer uXJx, Reviewer 3cbM). We appreciate these encouraging assessments and address remaining concerns below.

---

### Note · Authors · 2026-01-07

I have read and agree with the venue's withdrawal policy on behalf of myself and my co-authors.